# Novel Insights on Extracellular Electron Transfer Networks in the *Desulfovibrionaceae* Family: Unveiling the Potential Significance of Horizontal Gene Transfer

**DOI:** 10.3390/microorganisms12091796

**Published:** 2024-08-29

**Authors:** Valentina Gonzalez, Josefina Abarca-Hurtado, Alejandra Arancibia, Fernanda Claverías, Miguel R. Guevara, Roberto Orellana

**Affiliations:** 1Laboratorio de Biología Celular y Ecofisiología Microbiana, Facultad de Ciencias Naturales y Exactas, Universidad de Playa Ancha, Leopoldo Carvallo 270, Valparaíso 2360001, Chile; valentina.gonzalez.f10@gmail.com (V.G.); josefina.abarca@gmail.com (J.A.-H.); alejandra.arancibia@upla.cl (A.A.); 2Laboratorio de Microbiología Molecular y Biotecnología Ambiental, Departamento de Química & Centro de Biotecnología Daniel Alkalay-Lowitt, Universidad Técnica Federico Santa María, Avenida España 1680, Valparaíso 2390123, Chile; fclaveriasr@gmail.com; 3Departamento de Química y Medio Ambiente, Sede Viña del Mar, Universidad Técnica Federico Santa María, Avenida Federico Santa María 6090, Viña del Mar 2520000, Chile; 4HUB Ambiental UPLA, Universidad de Playa Ancha, Leopoldo Carvallo 207, Playa Ancha, Valparaíso 2340000, Chile; 5Laboratorio de Data Science, Facultad de Ingeniería, Universidad de Playa Ancha, Leopoldo Carvallo 270, Valparaíso 2340000, Chile; miguel.guevara@upla.cl; 6Núcleo Milenio BioGEM, Valparaíso 2390123, Chile

**Keywords:** sulfate-reducing bacteria, iron-reducing bacteria, extracellular electron transfer, mobilome

## Abstract

Some sulfate-reducing bacteria (SRB), mainly belonging to the *Desulfovibrionaceae* family, have evolved the capability to conserve energy through microbial extracellular electron transfer (EET), suggesting that this process may be more widespread than previously believed. While previous evidence has shown that mobile genetic elements drive the plasticity and evolution of SRB and iron-reducing bacteria (FeRB), few have investigated the shared molecular mechanisms related to EET. To address this, we analyzed the prevalence and abundance of EET elements and how they contributed to their differentiation among 42 members of the *Desulfovibrionaceae* family and 23 and 59 members of *Geobacteraceae* and *Shewanellaceae*, respectively. Proteins involved in EET, such as the cytochromes PpcA and CymA, the outer membrane protein OmpJ, and the iron–sulfur cluster-binding CbcT, exhibited widespread distribution within *Desulfovibrionaceae*. Some of these showed modular diversification. Additional evidence revealed that horizontal gene transfer was involved in the acquiring and losing of critical genes, increasing the diversification and plasticity between the three families. The results suggest that specific EET genes were widely disseminated through horizontal transfer, where some changes reflected environmental adaptations. These findings enhance our comprehension of the evolution and distribution of proteins involved in EET processes, shedding light on their role in iron and sulfur biogeochemical cycling.

## 1. Introduction

Sulfate- and iron-reducing prokaryotes (SRPs and FeRPs) are microorganisms that play a vital role in the biogeochemical cycles of sulfur and iron, two essential elements for the functioning of life on earth [1,2]. A vast body of research highlights the impact of the interaction of these two groups across many ecosystems, including anaerobic soils and sediments, pristine or contaminated freshwater, groundwater and marine environments, and even intestinal systems [3,4,5,6,7,8,9,10,11,12,13,14]. The interactions between both functional groups have been widely reviewed in marine environments, in which their activity accounts for most of the anaerobic organic matter degradation in sediments at the global level [15,16,17,18,19]. Currently, marine sulfate (~29 mM) is the largest mobile sulfur reservoir on our planet, an oxidizing pool even greater than atmospheric oxygen [20,21]. The mechanical and chemical weathering of continental rocks releases sulfur into the seawater column, which, in turn, continuously supplies sulfate to marine snow and sediments, where it is reduced by SRPs [16,20,22]. Analogously, iron enters the ocean from different sources, mainly remaining as a redox-active element in sediments, where FeRPs use it as an electron acceptor [3,23,24]. H_2_, formate, acetate, and other volatile fatty acids produced by hydrolysis or fermentation are used as electron donors for both dissimilatory processes, making the biogeochemical cycling of carbon, iron, and sulfur tightly linked [10].

Major determining factors contribute to the balance of how iron and sulfate reduction are spatially and temporally organized [25]. First, there is a competition for electron donors that was initially addressed by studying the concept of competitive exclusion [11,26,27,28,29,30]. Second, these biogeochemical processes are constrained to those metabolisms that include dissimilatory pathways capable of interacting with insoluble (iron) and soluble (sulfate) electron acceptors and by the free energy released by each redox reaction. While dissimilatory iron-reducing metabolisms, which demand that electrons must be effectively transported from cytoplasmic donors to extracellular space and generate higher free energy, dominate surface environments where Fe (III) is available, reductions in sulfate, yielding less energy, are restricted to deeper layers of sediments [3,23,31]. Third, several chemical reactions regulate the bioavailability or toxicity of both substrates and by-products. For instance, Fe(II) produced by dissimilatory iron reduction tends to diffuse to an oxic/anoxic interface, where it is reoxidized back to Fe(III) [23,32]. In contrast, Fe(III) works as an oxidant for sulfide that is produced by deeper SRPs, producing iron sulfide (FeS) and, lately, pyrite [33], which competes with the incorporation of sulfide into organic matter [34,35,36]. 

Based on this evidence, it was initially assumed that iron and sulfate reductions occur in discrete zones [27,37,38]. However, more recent observations have challenged this notion, revealing that both processes can coexist simultaneously [3,13,14,39] and even interplay along different stages of mineral transformation [40]. Phylogenetically diverse species of SRPs can reduce Fe(III) and Mn(IV) as well as electrodes of bioelectrochemical systems, suggesting that these capabilities may be widespread in various clades, although few of them can conserve energy to support growth [23,41,42,43,44,45,46,47,48]. Several SRP strains have been involved in the corrosion of Fe-containing metals by a combination of different mechanisms, suggesting many possible pathways of interactions with extracellular electron donors/acceptors [49,50]. In addition, comparative genomic studies have shed light on the lasting role that genetic mobile elements play in the plasticity and evolution of genomes of members of the *Desulfovibrionaceae*, *Geobacteraceae*, and *Shewanellaceae* families [51,52,53,54,55,56]. Recent investigations have reported that genes encoding transmembrane electron conduits in *Shewanella*, MtrCAB and OmcA, have been disseminated through horizontal gene transfer within the same species [57], genus [58], and across distantly related genera [59], suggesting that the mobilome may play a role in acquiring sophisticated metabolic processes such as extracellular electron transfer (EET). This juxtaposition led to the hypothesis that as a result of co-localization, collaboration, and competition, as well as their metabolic plasticity, several SRPs may have also evolved the capability to reduce insoluble Fe(III) oxides coupled with the oxidation of low-concentration electron donors.

The *Geobacteraceae* and *Shewanellaceae* families are FeRPs that are well known for their EET capabilities. The mechanisms of their model bacteria, *G. sulfurreducens* PCA and *S. oneidensis* MR-1, have been extensively studied by combining genomic, transcriptomic, and proteomic approaches coupled with functional genetic experiments, and they have been expanded to other strains [60,61,62,63,64,65]. Based on these well-studied mechanisms, we conducted a search to understand the prevalence of the orthologs of EET-related proteins in forty-two genomes of SRB belonging to the *Desulfovibrionaceae* family. The analysis included 130 proteins that encode *c*-type cytochromes, such as porin–cytochrome complexes (Pcc), *b*-type cytochrome complexes (Cbc), riboflavin biosynthesis genes, chemotaxis-related genes, and cell membrane components, and explored the mobilome of the three groups to assess the impact of horizontal gene transfer (HGT) on the acquisitions and losses of genes critical for EET by members of the *Desulfovibrionaceae* family.

Through phylogenetic and orthology analyses, we identified key proteins involved in EET that are widely distributed across these three families. Most of these proteins display modular diversification, serving as components of multiple complexes engaged in respiratory mechanisms. We believe this pool may include the essential core features required for the proper physiological functioning of EET and could constitute a crucial aspect of evaluating its expansion to other SRPs. These findings enhance our understanding of the evolution and distribution of proteins related to extracellular electron transfer processes, shedding light on their role in microbial communities actively participating in iron and sulfur biogeochemical cycles.

## 2. Materials and Methods

### 2.1. Genome Selection and Phylogenomic Analysis

A collection of already-sequenced genomes from both sulfate-reducing bacteria (SRB) and Fe-reducing bacteria (FeRB) were analyzed to investigate the shared genomic characteristics and the evolutionary relationships between the two groups. The selected genomes span 23 and 59 strains of the *Geobacteraceae* and *Shewanellaaceae* families, respectively, representing the FeRB, in addition to 42 genomes of SRB belonging to the *Desulfovibrionaceae* family (Appendix A). The sequence data for all of the bacterial genomes were retrieved from the NCBI RefSeq database (query date: March 2022). The quality of the genomes was analyzed with CheckM [66], using completeness > 98% and contamination < 5% as the cutting parameters. In order to perform a genome-wide phylogenetic analysis between all selected genomes, OrthoFinder [67] was used to perform a DIAMOND-based all-versus-all gene search on amino acid levels and identify clusters of orthologous genes (OGs). From this analysis, 109 single-copy OGs were aligned by MAFFT [68] and concatenated to construct a phylogenomic tree with FastTree [69] under the maximum-likelihood method. For the visualization of the tree, MEGA X [70] and iTOL v5 [71] were used.

### 2.2. Creation of Custom Databases: Genes Related to Extracellular Electron Transfer (EET)

To identify EET-related genes in the selected genomes, a custom database was generated using 130 gene sequences reported to be involved in EET from two known FeRB: *Geobacter sulfurreducens* PCA and *Shewanella oneidensis* MR-1 (Appendix A). This database includes several genes encoding *c*-type cytochromes, both outer-membrane and periplasmic; genes encoding porin–cytochrome complexes (Pcc) and *b*-type cytochrome complexes (Cbc); and genes related to riboflavin biosynthesis, chemotaxis, and cell membrane components, among others. The sequences were retrieved from the UniProt and NCBI NR databases (query date: January 2022).

### 2.3. Ecophysiological Analysis of SRB and FeRB

In order to relate and discover connections between the genomic and physiological characteristics of the bacteria analyzed, an extensive compilation of their ecophysiological information was carried out through a literature review, following and expanding what was previously described [53,72,73,74,75,76,77,78,79,80,81,82,83,84,85,86,87,88,89,90,91,92,93,94,95,96,97,98,99,100,101,102,103,104,105,106,107,108,109,110,111,112,113,114,115,116,117,118,119,120,121,122,123,124,125,126,127,128,129,130,131,132,133,134,135,136,137,138,139,140,141,142,143,144,145,146,147,148,149,150,151,152,153,154,155,156,157,158,159,160,161,162,163,164,165,166,167,168,169,170,171,172,173,174,175,176,177,178,179,180,181,182,183,184,185,186,187,188,189,190,191,192,193,194,195,196,197,198]. This analysis included the following parameters for each strain: (i) sources of isolation classified into the following categories: “Freshwater sediments”, “Brackish water/sediments”, “Marine water/sediments”, “Soil”, “Engineered/Impacted system”, “Plant/Algae-associated”, “Animal/Human-associated”, “Food”, and “Unknown”; (ii) growth conditions, such as pH and temperature growth ranges (with the terms “mesophile”, “psychrophile”, and “psychrotolerant” used as descriptors), and salinity tolerance, which was classified according to the maximum value of % NaCl in which growth was observed, or, if such information was not found, according to the source of isolation of each strain, classified as low (<1% NaCl, freshwater sediments/human-associated), medium (1–3% NaCl, brackish water/sediments/marine animal-associated), or high (>3% NaCl, marine water/sediments); and (iii) metabolic traits such as oxygen tolerance (with the terms Anaerobe, Facultative anaerobe, or Aerobe), electron acceptors and donors used for growth (using the terms complete or incomplete oxidation), growth rate estimations based on duplication capacity under optimum conditions (as fast-growing: in ≤12 h, or slow-growing: in >12 h), and iron reduction capacity. This information was later connected to phylogenomic and similarity network analyses.

### 2.4. Comparative Genomics

To investigate the shared genomic features related to EET between the SRB and FeRB, a search was performed for the OGs inferred by Orthofinder containing sequences from our custom database. The presence or absence of proteins from each strain was evaluated by determining whether or not the genome possessed the correspondent OG. The inferred OGs containing the EET-related sequences of each genome were sorted according to the phylogenomic tree to generate a heatmap on the R platform with the gplots package v.3.0.3. The detected proteins were subsequently analyzed to predict their cellular localization using the PSORTb web server [199].

Multi-heme *c*-type cytochromes (≥3 CXX(X)H motifs) were identified with a Perl script [200]. The subcellular localization prediction for genes containing heme motifs was performed using PSORTb v3.0.3 [199]. The predicted categories were extracellular, outer membrane, periplasmic, cytoplasmic membrane, and unknown. In addition, a search for prophage-like sequences was carried out in all the analyzed genomes using the PHAge Search Tool Enhanced Release (PHASTER) web server (http://phaster.ca/ (accessed on 1 June 2022)) with default parameters for closed and WGS data [201]. Histograms with this information were performed using GraphPad Prism (version 8.2.1), and incorporated into the phylogenomic tree. The presence of CRISPR elements in the closed and WGS data of each strain was analyzed with the CRISPRCasFinder web server (https://crisprcas.i2bc.paris-saclay.fr/ accessed on July 2022) using the default parameters [202]. To quantify the number and type of insertion sequences (ISs) present in the genomes of each strain, the web server of ISFinder was used (https://isfinder.biotoul.fr/ (accessed on 30 July 2023)), using the default parameters for a blastn and an E-value of 0.000001 [203]. The ISs were considered if they had an alignment length over 700 bp. For the detection of integrons in the closed and WGS data of each strain, the program IntegronFinder was used in the open platform Galaxy@Pasteur (https://galaxy.pasteur.fr/ (accessed from July to September 2023)), using the default parameters with the addition of the option of local detection and a search of promoter and attI sites [204]. The 3 types of elements recognized by IntegronFinder were complete integrons (integrons with an integron–integrase near *attC* site(s)), In0 elements (integron–integrase only, without any *attC* site nearby), and CALIN elements (cluster of *attC* sites lacking a nearby integrase). To detect the number of types of ICEs present in each strain (including the ones in the chromosomes and plasmids sequences if they were available), the CONJScan models in MacSyFinder were used [205,206]. This was individualized by checking the number of proteins and the types of conjugation system for each strain.

### 2.5. Similarity Network

To search for evidence of the extracellular electron transfer potential in the groups of bacteria analyzed, a similarity network analysis was performed using the sequences of potential cytochromes (sequences containing greater than or equal to three heme motifs) predicted as the extracellular or outer membrane, according to the analysis using the PSORTb web server. For this purpose, all protein sequences from the OGs that contained at least one extracellular or outer-membrane cytochrome were extracted and used to perform a BLAST all-vs-all analysis. The network was generated with 1807 putative cytochrome sequences belonging to the *Desulfovibrionaceae* (125), *Geobacteraceae* (918), and *Shewanellaceae* (764) families. Each node in the network represents a single protein sequence and each edge represents an alignment hit with an E-value of 10^−40^ or better. As filters for the network construction, an identity percentage of at least 30% and a minimum coverage of 70% were considered. The similarity networks were obtained using Gephi v.0.9.2 [207] with a combination of the Fruchtermann–Reingold [208] and Yifan Hu [209] layout algorithms. Community detection was performed using a weighted Louvain algorithm with a default resolution parameter of 1 [210].

### 2.6. Phylogenetic Analysis and Comparison of Genetic Contexts

The evolutionary relationship of the most prevalent EET-related genes shared between *Desulfovibrionaceae*, *Geobacteraceae*, and *Shewanellaceae* families was investigated using a phylogenetic analysis and the comparison of their genetic contexts. All the amino acid sequences of the OGs related to PpcA, OmpJ, CymA, CbcT, and CbcC were retrieved from the orthology analysis using Orthofinder. The gene trees of the selected OGs were recovered and visualized using iTOL v5 [71]. In addition, to facilitate the analysis of each OG, a sequence from each bacterium was chosen through local BLAST against the target protein, either from *Geobacter sulfurreducens* PCA or *Shewanella oneidensis* MR-1, and the visual comparison of their genetic contexts, giving greater priority to the latter. The selected amino acid sequences of each OG evaluated were aligned using MAFFT v7.511 [68]. The alignments were manually trimmed using the alignment editor AliView version 1.28 [211]. The phylogenetic reconstruction was determined by means of Bayesian Markov Chain Monte Carlo (MCMC) inference as implemented in MrBayes v3.2.7 [212]. Two independent runs were performed using a mixed amino acid substitution model where each run comprised 500,000 generations (two chains each run, sampling frequency of every 1000 generations). To construct the consensus tree, 25% of the trees were eliminated following a burn-in process. Posterior probabilities were used to support the internal branches. The visualization and editing of phylogenetic trees were performed using FigTree v. 1.4.4 software (http://tree.bio.ed.ac.uk/software/figtree/ accessed on 22 August 2024). Gene contexts were visualized using GeneSpy v1.2 [213], based on the GFF files from the NCBI RefSeq database.

### 2.7. Statistical Analysis

To understand how ecophysiological and genomic traits may influence the prevalence of elements related to microbial extracellular electron transfer (EET) in all the analyzed strains, we employed the statistical technique known as Principal Component Analysis (PCA). PCA is used for dimension reduction when dealing with numerous variables [214]. This analysis allowed us to use all the information provided by the various metrics without being biased by just a few aspects. Genomic traits (genome size and GC content), mobile genetic elements (number of prophages, CRISPR arrays, insertion sequences, integrons, and integrative and conjugative elements), and the number of copies of genes encoding proteins associated with EET-related elements (Cbc, CymA, Rib, OmpJ, and PpcA) were included in the analysis. Thus, PCA effectively splits genomes into groups reflecting both their sequence similarity and ecological distribution. PCA was performed using R software (version 4.3.1), selecting the first two principal components that captured most of the variance.

## 3. Results and Discussion

### 3.1. Phylogenomic Analysis of Sulfate-Reducing and Iron-Reducing Metabolism

To identify the shared genomic features between SRPs and FeRPs, a phylogenomic analysis including 124 genomes belonging to the *Desulfovibrionaceae*, *Geobacteraceae*, and *Shewanellaceae* families was reconstructed using 109 single-copy orthogroups (Figure 1). The analysis revealed that out of a total of 458,646 genes, 97.7% (448,320) were assigned to 15,156 orthologous groups (OGs). Of these, 289 OGs were present in all species, while 546 OGs were specific to certain species (Appendix A). The remaining unassigned genes and species-specific orthogroups represented the unique genetic traits of each species. It was also found that the *Desulfovibrionaceae*, *Geobacteraceae*, and *Shewanellaceae* families share 2372 species-shared OGs and have 3300, 1737, and 4707 specific OGs, respectively. Interestingly, the *Desulfovibrionaceae* family shares a greater number of exclusive OGs with the *Shewanellaceae* family (1301) compared to the *Geobacteraceae* family (1088) (Appendix A). 

The resulting phylogenomic reconstruction revealed three major clades, each corresponding to each family. The *Desulfovibrionaceae* clade includes 42 genomes, most of which have been isolated or recovered from pristine and contaminated ecosystems, as well as from a broad range of aquatic environments, from marine sediments to freshwater (Appendix A). The *Desulfovibrionaceae* clade contains strains with an average genome size of 3.9 Mbp and a G+C content of 60.8%, with the smallest genome corresponding to *Desulfovibrio piger* ATCC 29098 (2.87 Mbp) and the largest to *Desulfovibrio inopinatus* DSM 10711 (5.77 Mbp). The *Desulfovibrionaceae* clade is divided into two distinct subclades. Subclade I mainly comprises strains living in marine and estuarine ecosystems, whereas subclade II comprises strains from freshwater and engineered ecosystems (Appendix A). The *Geobacteraceae* clade includes 23 species of the *Geobacter* genus, a group of genomes belonging to species that have been isolated or recovered from soil and freshwater sediments as well as polluted sites, where *Geobacter* species play an important role in the regulation of biogeochemical cycles [215,216,217,218,219]. The *Geobacteraceae* clade contains strains with an average genome size of 4.0 Mbp and a G+C content of 58.3%. The smallest genome of this clade is *Geobacter benzoatilyticus* Jerry-YX (3.58 Mbp), and the largest genome is *Geobacter uraniireducens* Rf4 (5.14 Mbp). The *Geobacteraceae* clade is divided into two subclades. *Geobacter* strains isolated/found in soils, sediments, groundwater, and engineered environments belong to subclade I, while strains isolated from polluted sites and freshwater ecosystems belong to subclade II (Appendix A). The third clade comprises 59 species of the *Shewanella* genus, with genomes with remarkably low values of G+C content (45.3%) and an average genome size of 4.8 Mbp. The *Shewanella* clade includes genomes ranging from 3.9 Mbp to 6.4 Mbp of *Shewanella aestuarii* JCM 17801 and *Shewanella psychrophila* WP2, respectively. Unlike the *Desulfovibrionaceae* and *Geobacteraceae* clades, 56% (33) of the species belonging to this clade have been isolated or recovered from marine ecosystems, and secondly, from samples derived from soils and sediments (subclade I) as well as freshwater and human- and animal-associated environments (subclade II), where *Shewanella* has been recently found (Appendix A) [52,220]. 

As anticipated, our findings revealed variations in genomic traits, including GC content and genome size, among genomes from different subclades (Appendix A). In agreement with previous studies, the genomic GC content of strains of *Shewanellaceae*, a family within the class *Gammaproteobacteria*, was found to be lower than that of *Desulfovibrionaceae* and *Geobacteraceae*, which belong to *Deltaproteobacteria* [221]. Also, previous evidence has shown that genomes with higher GC content have more N in their proteomes [222]. Therefore, the lower GC content of *Shewanellaceae* strains may also be partially explained by the fact that several strains are primarily found in the ocean, an environment with a persistent limitation of N [223]. Another study reported that in the genome of *Desulfovibrio vulgaris*, mutations that convert GC to AT (GC->AT) were the most common, suggesting that a loss of GC content in this genome is slowly occurring [224].

In total, 34 strains out of the 124 surveyed had been implicated in some form of electron transfer to extracellular compounds, most of which belong to *Geobacteraceae* and *Shewanellaceae* clades. Both families have been the focus of a great extent of experimental evidence regarding their capabilities of extracellular electron transfer, which mainly relies on two types of mechanisms for electron transport across the outer surface. In *Shewanella* strains, substances that act as electron shuttles allow electrons to be transported from an intracellular enzymatic complex to the extracellular electron acceptor. This is the case with *Shewanella oneidensis* MR-1, which secretes small redox-active molecules for electron shuttling back and forth between cells and external electron acceptors [225]. In contrast, direct EET, which is prevalent in *Geobacter* strains, depends on the availability of redox-active enzymes and conductive appendages attached to the outer surfaces of the cells [226]. Four strains of the *Desulfovibrionaceae* clade, including *Maridesulfovibrio frigidus* DSM 17176 [46], *Desulfocurvibacter* africanus PCS [45], *Desulfovibrio vulgaris* str. Hildenborough [227], and *Desulfovibrio desulfuricans* DSM 642 [41], were shown to reduce Fe(III) and use it as an electron acceptor under experimental conditions. The extent to which this process takes place under environmentally relevant conditions as well as their molecular mechanisms remain to be explored. 

#### 3.1.1. Abundance of Multi-Heme Cytochromes

Multi-heme *c*-type cytochromes (*c*-Cyts) are proteins that harbor three or more hemes that have a central coordinated Fe atom that allows for the transfer of electrons. In species like *Shewanella oneidensis* MR-1 and *Geobacter sulfurreducens* PCA, *c*-Cyts play a fundamental role in EET to solid metal (hydro)oxides [228,229,230] as well as direct interspecies electron transfer [231,232]. *c*-Cyts are also very abundant in *Desulfovibrionaceae* [233]. To assess the diversity and prevalence of *c*-Cyts, we searched for the motifs CXXCH and CXXXCH across all strains, revealing a total of 9800 such proteins. On average, members of *Geobacteraceae* have 125.7 heme-containing proteins per genome, whereas *Shewanellaceae* and *Desulfovibrionaceae* have 77.1 and 56.1, respectively. The CXXCH motif was more common than the CXXXCH motif, representing between 84.4% and 94.3% of all proteins with heme motifs (Figure 2A). Interestingly, some *c*-Cyts exhibited both motifs, though this was more prevalent in *Geobacter* proteins (9.6% of proteins exhibited both motifs) and rare in *Shewanella*. *Geobacter* strains had an average of 10.6 extracellular predicted *c*-Cyts per genome, compared to 1.1 *c*-Cyts per genome of the *Shewanellaceae* family, and none of the *Desulfovibrionaceae* (Figure 2B). Regarding the number of motifs found per protein, most contain only one or two motifs. This ranges between 52.5% in the *Geobacteraceae* family and 77.5% in the *Desulfovibrionaceae* family. The abundance of multi-heme cytochromes in *Geobacteraceae* strains is particularly interesting, with an average of 59.7 per genome, which is significantly higher than other families, which contain 12.6 and 24.6 multi-heme proteins per strain, respectively (Figure 1 and Figure 2C). Some *Geobacter* strains, such as *G. uraniireducens* Rf4, *G.* sp. OR-1, *G. daltonii* FRC-32, and *G.* sp. DSM 9736, contain proteins with more than 40 heme motifs, whereas *Shewanella* and *Desulfovibrio* contain significantly less. In terms of cellular localization, it has been predicted that 9.5% of *Geobacter*’s multi-heme cytochromes and 0.7% of *Shewanella*’s are likely to be secreted from the cell. On the other hand, *Desulfovibrionaceae* strains do not seem to have extracellular multi-heme proteins or contain multi-heme proteins associated with the outer membrane (Figure 2). These findings agree with previous reports, where *Geobacter* species, such as *G. sulfurreducens*, encode many *c*-type cytochromes in their genomes compared to *Shewanella* and *Desulfovibrio* [64,233].

#### 3.1.2. Similarity Network Analysis of Extracellular Multi-Heme Cytochromes

A similarity network analysis was conducted to determine the phylogenetic relationships between the predicted multi-heme cytochromes located outside of cells. The analysis found 1807 sequences associated with 35 OGs (including some sequences that had not passed the extracellular localization filter). *Geobacter* strains had the majority of the sequences (918), followed by *Shewanella* strains (764) and *Desulfovibrionaceae* family strains (125). The similarity network had 130 sets of highly interconnected nodes (E-value threshold of 10^−40^), known as communities, with 71 containing two or more nodes (Figure 3). The network clusters exhibited a clade-specific pattern, indicating closer relationships based on their family of origin. Despite no clear correlation between the isolation source and clusters, it is evident that cluster proteins related to *Shewanella* strains are adapted to high-salinity conditions due to their origin from marine sources (Appendix A). Some cytochromes involved in EET, including OmcA and MtrC from *S. oneidensis* MR-1, and OmcA, OmcS, OmcZ, and CbcA from *G. sulfurreducens* PCA, are exclusively grouped with cytochromes of the same family. However, the OmcI cytochrome of *G. sulfurreducens* PCA, and the DsmE and MtrA cytochromes of *S. oneidensis* MR-1, were grouped in the same cluster along with other cytochromes of the *Shewanellaceae* and *Geobacteraceae* families. On the other hand, the *Desulfovibrionaceae* family presents nodes related to six clusters, three of which contain at least two cytochromes belonging to *Desulfovibrionaceae* strains capable of Fe reduction. The first of these clusters (community N#37, Figure 3B) is comprised exclusively of proteins of this family, whose products correspond to cytochrome *c* family proteins containing ten heme motifs. The second cluster (Figure 3C) contains 65 sequences (community N#33) encoding for a cytoplasmic membrane-bound cytochrome ubiquinol oxidase subunit I or *c*-type cytochromes. Both clusters related to OG 517 and OG 1638 also contain some *Geobacter* and *Shewanella* cytochromes predicted to be extracellular, and therefore, exploring their function in future studies may be interesting. 

#### 3.1.3. Comparative Genomic Analysis of Genes Related to Extracellular Electron Transfer Mechanisms

Since several molecular mechanisms for which members of the *Geobacteraceae* and *Shewanellaceae* families interact with extracellular electron acceptors have been widely described, we analyzed the dataset of *Desulfovibrio* genomes to learn about the ubiquity and abundance of proteins involved in EET (Figure 4). To conduct the analysis, we identified EET-related proteins in each genome by checking for their presence in the corresponding orthologous groups (OGs) from *S. oneidensis* MR-1 or *G. sulfurreducens* PCA. We evaluated 130 genes that encoded both outer-membrane and periplasmic *c*-type cytochromes, genes encoding porin–cytochrome complexes (Pcc) and *b*-type cytochrome complexes (Cbc), riboflavin biosynthesis genes, genes related to chemotaxis, and cell membrane components (Appendix A).

##### Similarities with the EET Mechanism of *G. sulfurreducens*

Investigations focused on *Geobacter* models, such as *G.sulfurreducens* and *G. metallireducens*, contributed to the comprehension of EET through multiple respiratory pathways [64,215,234]. These mechanisms mainly involve *c*- and *b*-type cytochromes in the inner membrane, ImcH and CbcL. The deletion of these genes impaired the ability to reduce electron acceptors with potentials above and below −0.1 V versus the standard hydrogen electrode (SHE) [235,236]. While both proteins have orthologs in all tested *Geobacter* species, only four *Desulfovibrionaceae* strains have CbcL homologs, including *M. frigidus* DSM 17176, a strain capable of Fe(III) reduction, but incapable of producing enough energy to support growth [46] (Figure 4). The genome of *G. sulfurreducens* contains four gene clusters encoding inner-membrane cytochromes, including Cbc3 (*cbcVWX*), Cbc4 (*cbcSTU*), Cbc5 (*cbcEDCBA*, where *cbcC* is also known as *omcQ*), and Cbc6 (*cbcMNOPQR*), that play a role in EET [64]. The deletions of *cbcV* and *cbcBA* resulted in a considerable decrease in Fe(III) reduction, and a transcriptional study found that cbcT was upregulated on insoluble metal oxides versus Fe(III) citrate [64,237,238]. All these gene clusters are conserved and widely distributed in all *Geobacter* species [239], except for the Cbc6 cluster, which is incomplete in seven strains, mainly belonging to *Geobacter* subclade I (Figure 5). The CbcOP subunits are CbcVW orthologous proteins from the Cbc3 cluster and are present in all *Geobacter* and *Shewanella* strains but only in a few strains of the *Desulfovibrionaceae* family. 

Among the periplasmic cytochromes, PpcA, MacA, and PccF have been characterized in more detail. It has been proposed that PpcA transfers electrons from the cytoplasmic membrane to the outer membrane, while MacA acts as a hydrogen peroxide reductase and transfers electrons to PpcA [240,241]. The expression of the gene encoding PccF was upregulated during growth on insoluble metal oxides, suggesting a possible role during EET [237]. Homologs of these three genes were heterogeneously distributed across the members of the three families. While several genes encoding PpcA were highly abundant in *Geobacter* (average of 4.9 genes per genome) and *Desulfovibrio* (average of 3 genes per genome), the gene encoding MacA was mostly shared between *Shewanella* and *Geobacter* strains (Appendix A). A porin–cytochrome complex (Pcc) capable of transferring electrons across a liposomal membrane is encoded by a periplasmic *c*-type cytochrome (OmaB/C), a porin-like protein (OmbB/C), and a reductase (OmcB/C). The Pcc protein complex reduces ferric citrate and ferrihydrite, similar to the MtrABC complex in *S. oneidensis* [242]. Three additional gene clusters encoding putative “electron conduits” involved in EET, including the porin–cytochrome (Pcc) complex extABCD, the porin–cytochrome (Pcc) complex extEFG, and the porin–cytochrome (Pcc) complex extHIJKL [243], were found to be highly prevalent in *Geobacter* species. However, no homologs were found in the *Shewanellaceae* and *Desulfovibrionaceae* family members (Figure 5). A similar distribution was found in the plethora of multi-heme *c*-Cyts associated with the outer membrane, including OmcS, OmcZ, OmcE, OmcT, and PgcA, which play different roles along the EET for both Fe (III) and Mn(IV) oxide reduction and electrode respiration (Figure 5 and Appendix A) [237,244,245,246,247,248,249]. In contrast, genes encoding the outer-membrane *c*-Cyts OmcI, a homolog of the CbcA subunit of *G. sulfurreducens*, and the outer-membrane protein J (ompJ), a channel known to influence the quantity and localization of cytochromes in the outer membrane [250], were found to be present in all the strains of *Geobacteraceae* and *Desulfovibrionaceae*, and partially in *Shewanellaceae* strains. Xap, an extracellular anchoring polysaccharide protein, has a crucial role in metal oxide attachment, cell–cell agglutination, and localization of essential *c*-Cyts. It possesses averages of 35, 29, and 23, high numbers of homologs, in the *Desulfovibrionaceae, Geobacteraceae*, and *Shewanellaceae* families, respectively [251]. Although recent evidence has highlighted the role of secreted riboflavins during DIET in *Geobacter* cocultures [252], this mechanism was not added to our analysis. 

##### Similarities with the EET Mechanism of *Shewanella oneidensis*

EET is mediated by CymA, a six multi-heme *c*-Cyts, in *S. oneidensis* MR-1. CymA oxidizes quinol in the cytoplasmic membrane and transfers electrons to Fcc3 and STC, which transport electrons to MtrA [253,254,255]. MtrA, MtrB, and MtrC form a trans-outer-membrane complex to transfer electrons to the bacterial surface. MtrC and OmcA can physically interact with each other and transfer electrons directly to Fe(III) minerals [256,257,258], as well as associate with extracellular structures that were previously referred to as ‘nanowires’ [259]. *S. oneidensis* MR-1 employs endogenously produced flavin electron shuttles to enhance EET to minerals and electrodes during anaerobic respiration. Thus, released flavins are proposed to function as diffusive electron shuttles that transport electrons from MtrC and OmcA to mineral surfaces [225,260,261]. Homologs of CymA were found ubiquitously in all three families analyzed, with the exception of genomes of strains belonging to *Geobacter* subcluster I. In contrast, homologs of OmcA were found to be distributed in almost all the species of *Shewanella* and a few of *Geobacter* (Figure 5). Whereas Fcc3 homologs were found mainly in *Shewanella* strains, homologs of cctA, the gene coding the tetraheme STC, were shared by *Shewanella* strains and all strains of *Desulfovibrio desulfuricans* (Figure 5 and Appendix A). In addition to that, one guanosine triphosphate (GTP) and two ribulose-5-phosphate molecules are converted into one riboflavin molecule in a stepwise manner by the enzymes encoded by the ribA, ribB, ribD, ribH, and ribE genes [262]. This pathway seems to be ubiquitous in *Shewanella* species, although homologs of some of these genes, specifically ribB, ribD, and ribH, are also found in all species belonging to the *Geobacteraceae* and *Desulfovibrionaceae* families, which could imply some role of these molecules in their EET mechanisms. Further experimental research is required to investigate if flavins play a relevant role during EET by SRPs, as the addition of riboflavin and flavin adenine dinucleotide (FAD) showed the accelerated corrosion of carbon steel and stainless steel by *D. vulgaris* [263,264]. 

### 3.2. Mobilome Analyses across Members of Desulfovibrionaceae, Geobacteraceae, and Shewanellaceae Families

The genetic makeup of prokaryotic genomes is composed of DNA fragments from both vertical and horizontal gene transmission. The mobilome, a collection of mobile genetic elements, facilitates the transfer of genes and their corresponding functions within a community through horizontal gene transfer (HGT) [265]. Our analysis reveals that these SRP and FeRP families host various mobile genetic elements, such as plasmids, bacteriophages, integrons, insertion sequences (ISs), and integrative and conjugative elements (ICEs). According to a PHASTER search, 333 prophages were found in total, with 59 intact, 244 incomplete, and 30 questionable phages present in 122 strains (Figure 6A). The average number of prophages per strain was 3.1, 2.7, and 2.5 in *Desulfovibrionaceae*, *Geobacteraceae*, and *Shewanellaceae*, respectively. This variation is influenced by a combination of genomic, phenotypic, and environmental factors, including genome size, physiological status, and the specific habitat in which the strain resides [266,267,268]. Our findings revealed that strains from *Desulfovibrio* subclade II and *Geobacter* subclade I, predominantly present in soils, freshwater, subsurface, and engineered ecosystems, exhibited the highest prophage density (prophages per Mbp genome). In contrast, strains from *Shewanella* subclade I and *Desulfovibrio* subclade I, more prevalent in marine environments, displayed the lowest values (Appendix A). Clustered regularly interspaced short palindromic repeats (CRISPR) is a system that allows the identification and cleavage of foreign DNA. The presence of CRISPR-Cas arrays constitutes a barrier to HGT, including natural transformation, transduction and conjugation [269,270,271]. Our analysis revealed a high prevalence of CRISPR arrays in the genomes of *Desulfovibrionaceae* and *Geobacteraceae*, while a significantly lower prevalence (less than 30%) was observed for *Shewanellaceae* strains (Figure 6B). This trend could be explained by the recent identification of phages with genes encoding proteins capable of inhibiting CRISPR-Cas function in several *Shewanella* strains [272,273,274]. Integrons are genetic units that capture genetic material in bacteria, adding novel features to the cell that contains it. They consist of an integron–integrase gene, an integration site, a promoter for gene cassettes, and up to 200 gene cassettes containing open reading frames flanked by *attC* recombination sites [275,276]. We found complete integrons in over 50% of strains of *Shewanellaceae* and subclade II of *Geobacteraceae*, while they were absent in the other clades. A similar pattern was found for the prevalence of insertion sequences (ISs), which are cryptic DNA segments containing passenger genes that contribute to the metabolic plasticity and evolution of microbial genomes [277,278]. Our results indicate that the prevalence and number of ISs are unevenly distributed across the different bacterial groups included in this study. The *Shewanella* and *Geobacter* genomes contained 584 (~10.8 IS per genome) and 234 (~13 IS per genome), respectively, while the *Desulfovibrio* genomes contained a total of 34 ISs (~2.3 ISs per genome), with 27 out of the 42 *Desulfovibrio* showing no detection of ISs (Appendix A). With few exceptions, including *Desulfovibrio vulgaris* str. Hildenborough, the low prevalence of ISs in *Desulfovibrio* species is consistent with previous studies, suggesting limited genomic rearrangements by transposition in this genus [279,280]. This may be explained by the fact that some of the ISs found to belong to families (i.e., IS*Dvu3*) in which the control of transposase expression relies on stop codon read-through and, therefore, may be affected by other regulatory mechanisms [281]. Between 35 and 61% of genomes were found to contain integrative and conjugative elements (ICEs). In agreement with previous reports and in contrast to what was found with integrons and ISs, strains belonging to Shewanellaceae registered a lower prevalence of ICEs than the other two families [52]. 

The data from 124 genomes were used to conduct a Principal Component Analysis (PCA) of thirteen genomic and mobilome variables and metrics reflecting the prevalence of EET elements (Figure 7). The analysis clustered the strains into three big groups on the PC1-PC2 plane (accounting for 50.88% of the total data variability), revealing a stronger correlation with the taxonomy of species than with their habitats (Appendix A). Based on the number of copies of genes encoding CymA, PpcA, and riboflavins, the number of CRISPR arrays and integrons per genome, genome size, and GC content, the *Shewanellacea* group is farther from the *Geobacteraccea* and *Desulfovibrionaceae* groups. The latest two groups are separated along the PC2, which includes the number of total cytochromes and genes encoding OmpJ and Cbc-related genes, as well as PpcA, which is almost absent in *Shewanellaceae* strains (Figure 5, Appendix A). Interestingly, the number of copies of genes encoding proteins related to EET contributes to the differentiation within the three groups. While the number of copies of genes encoding for CymA and riboflavins contributed to the distinction between *Shewanella*, the number of copies of genes encoding for OmpJ contributed to the distinction of the *Desulfovibrionaceae*. In a similar manner, the number of copies of genes encoding for Cbc and total cytochromes contributed to the distinction of the *Geobacteraceae* and the other two groups. Thus, this result suggests that the prevalence and abundance of EET elements significantly contributed to the differentiation of these groups. 

### 3.3. Evolutionary Relationship of the Most Prevalent EET-Related Genes in SRPs

Many of the genomes belonging to the *Desulfovibrionaceae* family were found to possess genes homologous to crucial proteins involved in the EET mechanisms of *S. oneidensis* MR-1 and *G. sulfurreducens.* This includes the triheme periplasmic cytochrome PpcA, the outer-membrane protein OmpJ, the tetraheme *c*-type cytochrome CymA, the iron–sulfur cluster-binding protein CbcT (a subunit of the Cbc4 complex), and, to a lesser extent, the cytochrome *c*-type CbcC (a subunit of the putative Cbc5 complex). All of these exhibited widespread distribution within this family. To gain insights into the evolutionary relationship of these shared genes among the *Desulfovibrionaceae*, *Geobacteraceae*, and *Shewanellaceae* families, we conducted phylogenetic analysis and compared the genetic contexts of the select genes associated with the orthologous groups of these candidate proteins. 

#### 3.3.1. The Periplasmic Cytochrome PpcA, an Intermediary in Extracellular Electron Transfer

PpcA is a periplasmic cytochrome that acts as an intermediary electron carrier for EET. Genetic studies have found that PpcA acts as a terminal reductase for anthraquinone-2,6-disulfonate (AQDS), Fe(III)-citrate, and Ferric nitrilotriacetate (Fe-NTA), although this gene was not differentially expressed when *G. sulfurreducens* was grown with Fe(III) citrate and Fe(III) oxide [240,282,283]. The genetic context of *ppcA* in *G. sulfurreducens* includes several adjacent genes encoding *c*-type cytochromes, as well as genes involved in their biogenesis, such as ResB and CcsB [284], and genes involved in the biosynthesis of menaquinones and ubiquinones, redox-active compounds involved in respiratory networks [285,286] (Appendix A). The genomic context remains largely invariant across different species, and its phylogenetic relationship aligns with the species’ phylogenomic tree, suggesting vertical transmission rather than horizontal gene transfer. 

PpcA is part of a family of five periplasmic triheme cytochromes (including PpcB, PpcC, PpcD, and PpcE). Thermodynamic characterization of those cytochromes revealed differences in their heme reduction potentials, allowing for a wider range of redox partners and enhancing the adaptability of the respiratory mechanism [287,288]. Our findings revealed that ppcA homologs are prevalent in *Geobacter* species, with most strains containing between 5 and 6 homologous genes, averaging 4.9 genes per strain. Notably, ppcA homologs were also present in all examined members of the *Desulfovibrionaceae* family, with the majority of those having between 3 and 4 homologous genes (averaging about 3.4 genes per strain) (Appendix A). 

The phylogenomic analysis identified four primary clades within the PpcA protein family (Figure 8). Clade 1 exclusively consists of proteins from *Geobacter* strains. Notably, the five known homologous proteins from *G. sulfurreducens* (PpcA, PpcB, PpcC, PpcD, and PpcE) are distributed across different subclades within this group. Proteins similar to PpcA from the *Desulfovibrionaceae* family, found in the remaining three clades, exhibit significant diversification within their respective subclades. Similar to *Geobacter*, the presence of multiple variants and their diversification is likely the result of functional diversification associated with heme reduction. One example of evolutionary divergence is the gene encoding for the PpcA protein of *Pseudodesulfovibrio mercurii* (WP_014320801.1), which, based on genomic context, falls outside the four main clades (Appendix A). The unique environmental characteristics of the mid-Chesapeake Bay estuarine sediments where *P.mercurii* was isolated, including complex geochemical processes, nutrient-rich reducing waters, and the presence of rare earth elements (REEs), such as Cerium (Ce) and Europium (Eu), might have been significant factors driving this divergence [289,290]. In contrast, homologs of *ppcA* in two *Shewanella* strains, *S. atlantica* HAW-EB5 and *S. sediminis* HAW-EB3, both isolated from marine sediments near Halifax Harbour, Canada [291,292] suggest a horizontal gene transfer (HGT) event from another bacterium in this geographical region. Furthermore, clade 2 mainly comprises homologous proteins from members of the *Desulfovibrionaceae*, except for a PpcA homologous from *Geobacter* sp. SVR (WP_239077329.1), suggesting potential horizontal gene transfer events based on an interruption in the phylogenetic tree topology. These assumptions are supported by the presence of transposase-encoding genes in the vicinity of these genes [293,294].

#### 3.3.2. OmpJ, an Integral and Widespread Outer Membrane Protein in the Desulfovibrionaceae Family

Homologs of OmpJ were found to be widely distributed in the *Desulfovibrionaceae* family. Similar to *Geobacter* species, most of these strains have between one and two homologous ompJ genes. Interestingly, *Desulfovibrio desulfuricans* strains and those from the *Solidesulfovibrio* genus stand out, with most of their members having between eight and ten genes homologous to ompJ (Appendix A). This substantial prevalence of OmpJ-like genes suggests that this protein may have a role in the physiology, adaptation, and capabilities of participating in sulfate reduction and metallo-reduction processes [41].

The distribution of the ompJ phylogenetic tree shows three clear groups: one from *Geobacter* and two from the *Desulfovibrionaceae* family (Appendix A). The two branches of the *Desulfovibrionaceae* family consist mainly of species isolated from marine environments, pollution events, or industrial activity, while the other group primarily comprises species isolated from soil, freshwater, or animal sources (Appendix A). One exception is *Halodesulfovibrio aestuarii* DSM 17919, isolated from shoal mud in Germany [295], which stands out as it harbors two similar proteins that are significantly different from the other genes. This discrepancy in the species tree may suggest a horizontal transmission event, since near these ompJ-homologous genes, several tRNA sequences are found. To date, there have been limited reports regarding the function of OmpJ. Therefore, it would be of interest to assess its potential role in signaling mechanisms and its impact on EET mechanisms.

#### 3.3.3. CymA, a Common Branch Point in the Electron Transport Chain

The *c*-type cytochrome CymA seems to be essential for facilitating the anaerobic respiratory adaptability of *Shewanella*. CymA plays a key role in transferring electrons from menaquinol to various systems responsible for reducing terminal electron acceptors, such as fumarate, nitrate, nitrite, dimethyl sulfoxide (DMSO), arsenate, and insoluble minerals like Fe(III) and Mn(IV) [253,296,297,298]. Homologous genes to cymA are widespread among the members of the analyzed families. The phylogenetic tree of CymA reveals three main clades: two from *Shewanella* strains and a third shared between strains of *Geobacter* and the *Desulfovibrionaceae* family. While *Shewanella* species contain between 1 and 5 homologs (with an average of 2.7), almost all members of the *Desulfovibrionaceae* family contain 1 or 2 homologs, with a few exceptions. Nearly half of the species analyzed in *Geobacteraceae* contain one to two genes homologous to cymA (Appendix A). The variable presence of these genes in *Geobacteraceae* species suggests two potential evolutionary scenarios: the genes were either lost in most species or inserted into the genomes of the analyzed species. Notably, species containing these genes are mostly clustered within a specific clade in the phylogenetic tree (Appendix A), hinting at the likely insertion of this gene through horizontal gene transfer into the common ancestor of these species, followed by its subsequent vertical transmission. Another notable sequence in the phylogenetic tree is one of the homologous copies from *Geobacter hydrogenophilus* DSM 13691 (WP_214187890.1), positioned between two subclades of *Desulfovibrionaceae* family proteins (Appendix A). The context of this sequence suggests it was likely acquired via horizontal gene transfer, especially considering the proximity of genes encoding site-specific integrases and recombinase family proteins. Interestingly, when investigating the gene contexts of cymA homologs, several members of the *Desulfovibrionaceae* and *Geobacteraceae* families feature one of these homologous genes located near the gene encoding ammonia-forming cytochrome c nitrite reductase subunit c552 (Appendix A). The similarities between the genetic contexts of both families, along with their closer phylogenetic proximity in comparison to *Shewanellaceae* species (Appendix A), suggest that the potential insertion into *Geobacteraceae* genomes might have originated from horizontal transmission from a member of the *Desulfovibrionaceae* family or closely related species. 

#### 3.3.4. Inner-Membrane Quinone Oxidoreductase Protein Complexes: CbcC and CbcT Subunits Provide Plasticity and Modularity to Different Complexes Involved in EET

Cytochrome bc1 complexes are membrane protein complexes found in the electron transfer chains of bacteria using oxygen, nitrogen, and sulfur compounds as electron acceptors. These enzymes transfer electrons from ubiquinol to cytochrome c and move protons across the membrane. Despite transcriptomic and proteomic studies revealing differential expression patterns of Cbc-like gene clusters in *G. sulfurreducens* in response to electron acceptor availability, there is still limited information available regarding these complexes [237,244,299,300].

CbcT homologs are present in high abundance across all three families. In the *Desulfovibrionaceae* family, which has the highest average number of cbcT homologs (9.8), *Pseudodesulfovibrio mercurii* is noteworthy with 17 copies (Appendix A). *G. uraniireducens* Rf4 and *S. sediminis* HAW-EB3, from the *Geobacteraceae* and *Shewanellaceae* families, respectively, stand out with the largest numbers of CbcT orthologs, at 11 and 19 genes, respectively, except for *S. denitrificans*, which does not contain these genes. The intricate topology of the phylogenetic tree mirrors the abundance of this gene. Specifically, distinguishing clades by family is challenging due to their phylogenetically interwoven sequences. Moreover, there does not appear to be a relationship between abundance and sources of isolation, as the amount varies across all environmental classifications (Appendix A).

The cbcSTU operon is highly conserved among *Geobacteraceae* species (Appendix A). Among the strains analyzed, only *Geobacter* sp. FeAm09 presents homologs of cbcT, but neither cbcS not cbcU. Several *Shewanellaceae* strains exhibit a closely related cluster, featuring an orthologous of cbcT, which is a homolog to the sirC gene in *S. oneidensis* MR-1. SirC is a 4Fe-4S ferredoxin that, together with its partner SirD, encoding an NrfD/PsrC-type quinol dehydrogenase, has the ability to transfer electrons from quinols to the same respiratory pathways as CymA, except for nitrate. As a consequence, this quinol dehydrogenase complex (SirCD) can functionally replace CymA in the respiratory pathways for fumarate, DMSO, and ferric citrate as the electron acceptor [301]. It is worth noting that in most of the *Shewanellaceae* strains, there are two genes adjacent to one of the homologous cbcT genes. These two genes do not have homology to cbcS and cbcU but share the same annotation as the latter two genes, and are arranged in a similar spatial disposition. Specifically, the gene encoding the cytochrome c3 family protein (similar to CbcS) is homologous to the outer-membrane lipoprotein *c*-type cytochrome OmcI in *G. sulfurreducens* PCA. In this case, similar to the cbcU gene in *G. sulfurreducens*, there is a gene encoding a cytochrome c nitrite reductase subunit NrfD, although it does not belong to the CbcU orthogroup. These observations suggest a convergent evolution event in forming these modular membrane complexes, which appear to function similarly in electron transfer from the quinol pool in different respiratory pathways [302,303]. Interestingly, *S. sediminis* HAW-EB3, which presents the greatest number of cbcT homologs (and which acquired a ppcA homolog), also presents a high number of copies of other genes involved in EET mechanisms, including cbcA/omcI (12) and mrtA (12) homologs. This strain, isolated from an unexploded ordnance dumping site in the Atlantic Ocean near Halifax Harbour, Nova Scotia, Canada, can degrade RDX, nitrate, and nitrite. However, it does not demonstrate a reduction in Fe(III) or elemental sulfur [291]. These distinctive characteristics are likely a result of the environmental pressures in its natural habitat, which gives it unique features compared to other species. Furthermore, these observations suggest that these proteins are not limited to iron- or sulfur-reduction pathways from natural sources, but to other compounds, including chemicals from industrial activity and pollution events, to which bacteria have had to adapt. Considering its genetic features, it would be interesting to conduct a more in-depth characterization to investigate the functional potential of these mechanisms. In contrast, *S. violacea* DSS12 and *S. denitrificans* OS217 either lack or have only one homolog of cbcT. These strains also lack other crucial genes involved in EET mechanisms, such as OmcA, CctA, FccA, and the MtrCAB complexes. Recently, Baker et al. (2021) also reported the absence of this last complex in both species and attributed it to an environmental pressure effect, which was linked to the transition to an aerobic environment, facilitated by their habitat at the oxygenated sediment-water interface [59,304,305]

Regarding the *Desulfovibrionaceae* family, 24 of the 42 strains analyzed contain at least one homolog of cbcC. This distinct group forms a well-differentiated clade that exhibits significant evolutionary divergence from *Geobacteraceae* and *Shewanellaceae*, suggesting adaptation to the metabolic needs of these bacteria since their last common ancestor. In fact, this group of cbcC homologs coincides with one of the clusters of the multi-heme cytochrome similarity network (Figure 3B), specifically community N#37. This cluster comprises cytochrome *c* family proteins with 10 heme motifs, forming a separate cluster from other nodes in the network, thereby supporting the observed evolutionary divergence in the phylogenetic tree. Notably, in these species, at least one of the cbcC homologs was found alongside the rnfABCDGE operon, which was initially identified in *Rhodobacter capsulatus* [306]. The RNF complex is composed of six subunits, including four membrane proteins (RnfA, RnfD, RnfE, and RnfG) and two iron–sulfur proteins (RnfB and RnfC), and encodes a membrane-bound NADH:ferredoxin dehydrogenase [306]. Our results revealed that many members of the *Desulfovibrionaceae* family exhibit the loss of the rnfB gene and instead have a gene encoding an FAD-dependent oxidoreductase at the end of the operon (Figure 9). In the Bacteroidota/Chlorobiota group, the loss of the rnfB gene has been documented, along with the recruitment of a reductase subunit from aromatic monooxygenases (AMOr protein), resulting in the emergence of the sodium-dependent NADH:ubiquinone oxidoreductase (Na^+^-NQR). This complex is commonly associated with the aerobic respiratory metabolism of pathogenic bacteria. A key distinction between Rnf and Na^+^-NQR is the mechanism of electron incorporation into the complex, suggesting an alternative mechanism for electron transfer in the presence of this oxidoreductase in the Rnf complex in *Desulfovibrionaceae* family strains [307]. The presence of this oxidoreductase in the Rnf complex in strains of the *Desulfovibrionaceae* family may indicate an alternative mechanism for electron transfer. Interestingly, the strains exhibiting this substitution are typically found in marine or impacted environments, which are known for their harsh conditions compared to the habitats of strains with the RnfB subunit (mainly found in soil, animals, and freshwater). These findings imply that environmental factors such as oxygen levels and the presence of metallic and organic electron acceptors or donors may have driven the modification of the rnf operon.

It is important to note that in almost all strains of the *Desulfovibrionaceae* family that do not have *cbcC* homologs, the other subunits of the Rnf complex are also missing. This supports a hypothesis regarding the integration of the CbcC homolog as an accessory protein within the complex. The only exceptions are *Desulfovibrio cuneatus* DSM 11391, which possesses the *rnf* operon minus the RnfB subunit and the CbcC homolog, and *Halodesulfovibrio aestuarii* DSM 17919, which harbors the complete *rnf* operon and a cytochrome c, not homologous to CbcC (Appendix A). All other strains with the *rnf* operon contain the *cbcC* homolog, likely indicating an ancient evolutionary incorporation event before these species diverged. Furthermore, the sporadic occurrence of this complex in some family members strongly suggests horizontal gene transfer as a mode of acquisition. Previous research indeed indicates the spread of this complex among various lineages by HGT, including several species within the phyla *Pseudomonadota*, *Chlamydiota*, and *Planctomycetota*, which subsequently led to the rise in other complexes, such as Na^+^-NQR mentioned above [308]. In this context, the CbcC-like cytochrome represents a unique module that has become integrated into various membrane complexes involved in electron transfer through evolution and species diversification. In *Geobacter*, it is part of the CbcEDCBA cluster where it is predicted to form menaquinol:ferricytochrome c oxidoreductase [237], and in *Desulfovibrionaceae* family members, it forms part of the Rnf complex, with a predicted cytoplasmic location, where it likely engages in coupling to facilitate efficient electron transfer (Figure 9). These findings indicate that this subunit, adapted by *Desulfovibrionaceae* family members with significant evolutionary divergence, might offer a potential new catalytic innovation. By incorporating it into the Rnf complex, these bacteria could potentially broaden the detection of redox potentials and access an alternative electron transfer pathway, thereby enhancing growth efficiency in variable environments.

## 4. Implications

The coexistence of sulfate- and iron-reducing bacteria can be found in several environments with limited oxygen availability and varying redox conditions, such as marine sediments, anoxic soils, and groundwater. These bacteria have developed mechanisms for utilizing external electron donors and acceptors for energy metabolism, including EET. However, the distribution, diversity, and evolution of the EET mechanisms are still not well understood, and the underlying molecular mechanisms shared between SRB and FeRB remain unexplored. In this study, a comparative genomic analysis uncovered the similarities and differences in the distribution of genes related to EET in the genomes of the *Desulfovibrionaceae*, *Geobacteraceae*, and *Shewanellaceae* families. Our results showed a higher abundance of multi-heme cytochromes, especially those that were extracellular, in *Geobacteraceae* strains than *Desulfovibrionaceae* and *Shewanellaceae.* The analysis also showed more orthologous groups (OGs) shared between these two families than *Geobacteraceae*, suggesting a closer phylogenomic relationship. However, this did not necessarily correlate with OGs related to EET. In fact, the strains belonging to *Desulfovibrionaceae* shared more homologous genes related to EET with the model *G. sulfurreducens* PCA compared to *S. oneidensis* MR-1. 

Within each family, a set of genes related to EET proteins exhibited significant enrichment. For example, the presence of gene copies encoding CymA and riboflavins distinguished *Shewanellaceae*, while those encoding OmpJ differentiated the *Desulfovibrionaceae*. Similarly, the presence of gene copies encoding Cbc and total cytochromes distinguished the *Geobacteraceae* from other families. After conducting a comprehensive analysis, we identified potential horizontal gene transfer events, gene gain or loss events, and instances of convergent evolution related to specific proteins. These findings enhance our understanding of the distribution and evolution of EET genes and pathways across diverse phylogenetic groups of SRB and FeRB. These discoveries contribute to our understanding of the adaptability of these bacteria and their diverse electron transfer pathways in different environments, especially relevant in environments with redox gradients.

## Figures and Tables

**Figure 1 microorganisms-12-01796-f001:**
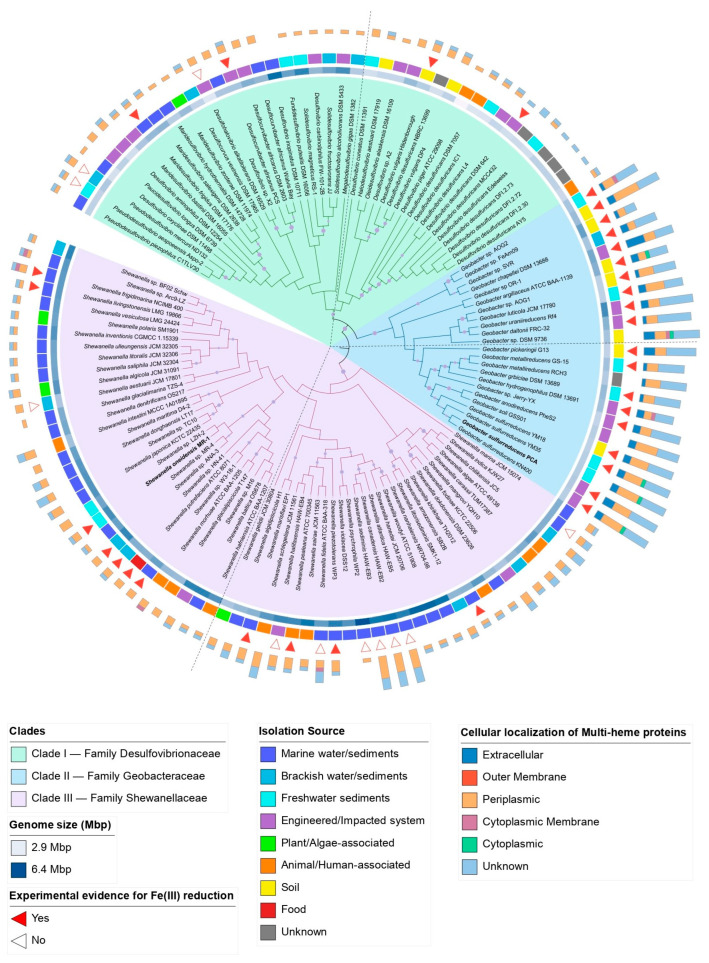
Phylogenomic cladogram of the analyzed genomes belonging to the *Desulfovibrionaceae*, *Geobacteraceae*, and *Shewanellaceae* families. Phylogeny was inferred using Orthofinder v2.5.4, identifying 109 single-copy orthogroups with all species present. Clades of each family are shown in colors. Blue scale rectangles beside the strains indicate a genome size from 2.9 to 6.4 Mbp. The adjacent squares represent the isolation sources. Red and white triangles indicate strains capable of/incapable of Fe(III) reduction in experimental assays. The absence of triangles indicates that no studies are available. The stacked bar graph shows the number of multi-heme *c*-type cytochromes (≥3 Cxx(x) H motifs) of each strain classified according to its cellular localization. *Geobacter sulfurreducens* PCA and *Shewanella oneidensis* MR-1 strains are shown in bold font. Bootstraps between 0.5 and 1 are indicated as purple circles of variable diameter in the respective node.

**Figure 2 microorganisms-12-01796-f002:**
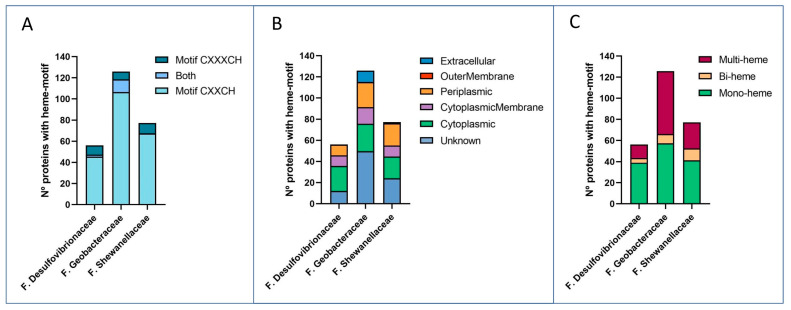
The abundance and diversity of proteins containing heme motifs in the *Desulfovibrionaceae*, *Geobacteraceae* and *Shewanellaceae* families. The average number of proteins containing heme motifs in the strains of each family, classified according to (**A**) type of heme motif, CXXCH, CXXXCH, or with both motifs; (**B**) cell localization predicted by the PSORTb web server, and (**C**) the number of motifs: mono-heme (1), bi-heme (2), and multi-heme (≥3).

**Figure 3 microorganisms-12-01796-f003:**
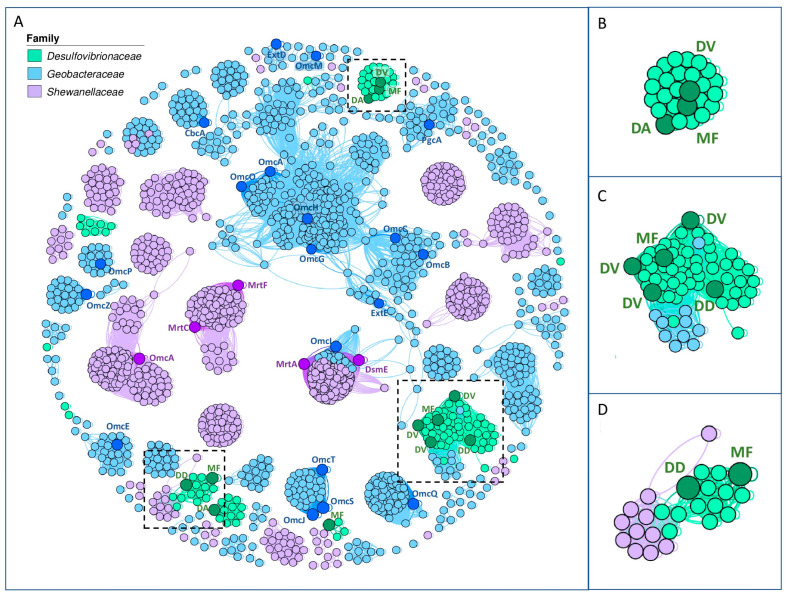
Multi-heme cytochrome similarity network. The distance network was constructed based on the protein sequences present in OGs containing multi-heme cytochromes (1807 sequences) predicted as extracellular from the 124 genomes of *Desulfovibrionaceae* (125 sequences), *Geobacteraceae* (918 sequences), and *Shewanellaceae* (764 sequences) families. The E-value threshold of the blast alignment for the network is 10^−40^. Each node represents an extracellular multi-heme cytochrome, and the color fill indicates the origin of the sequence. The family *Desulfovibrionaceae* is green, the family *Geobacteraceae* is blue, and the family *Shewanellaceae* is violet. The nodes highlighted in a larger and more intense color show previously studied cytochromes of known metal-reducing bacteria, in blue for *G. sulfurreducens* PCA and violet for *S. oneidensis* MR-1. Green highlights the cytochromes of the bacteria *Desulfocurvibacter* africanus PCS, *Desulfovibrio desulfuricans* DSM 642, *Desulfovibrio vulgaris* str. Hildenborough, and *Maridesulfovibrio frigidus* DSM 17176, which have also been reported to show Fe(III) reduction. These cytochromes are highlighted by their initial letters, DA, DD, DV, and MF. (**A**) Entire network. (**B**–**D**) Zooming into particular network clusters possessing at least two cytochromes of the metal-reducing bacteria of the *Desulfovibrionaceae* family mentioned above.

**Figure 4 microorganisms-12-01796-f004:**
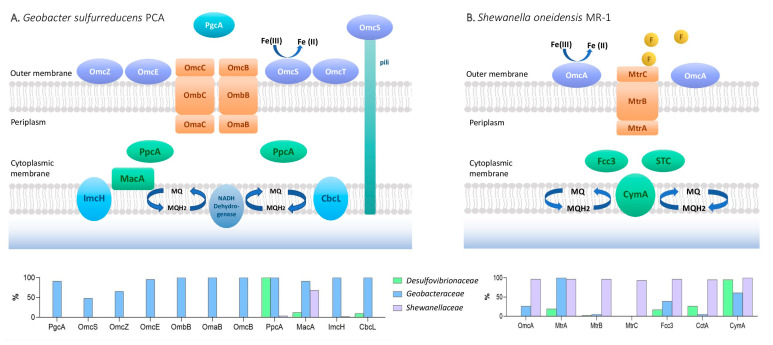
Proposed mechanisms for bacterial EET in model FeRB. The figures represent the mechanisms of *Geobacter sulfurreducens* PCA and *Shewanella oneidensis* MR-1. At the bottom, the bar plots show the percentage of the genomes in a given family that encode each EET-related gene.

**Figure 5 microorganisms-12-01796-f005:**
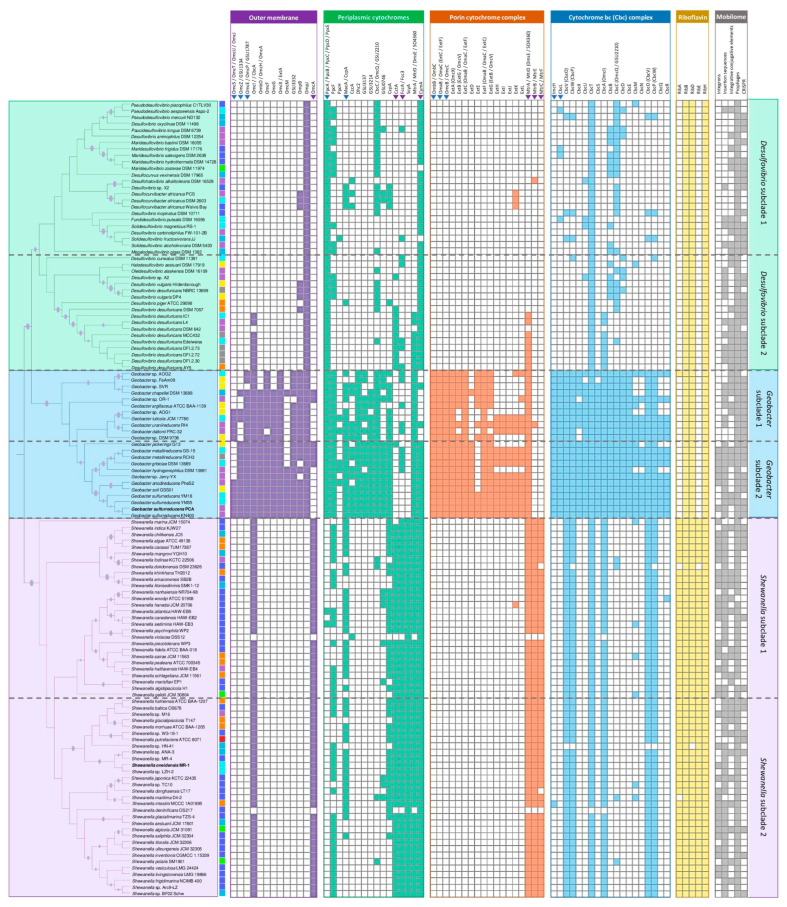
The presence/absence matrix of genes in OGs related to the EET mechanisms of *S. oneidensis* MR-1 y *G. sulfurreducens* PCA. On the left, the phylogenomic cladogram is presented with the isolation source. Boxes indicating the presence or absence of genes in the OGs involved in EET mechanisms are colored according to cell localization as follows: purple for outer-membrane cytochromes/proteins, green for cytochromes and proteins located in the periplasm, orange for porin–cytochrome complexes, blue for cytochrome bc (Cbc) complexes, and yellow for genes involved in riboflavin biosynthesis. The blue and purple triangles on the heatmap report the relevant genes in the EET mechanisms of *G. sulfurreducens* PCA and *S. oneidensis* MR-1, respectively.

**Figure 6 microorganisms-12-01796-f006:**
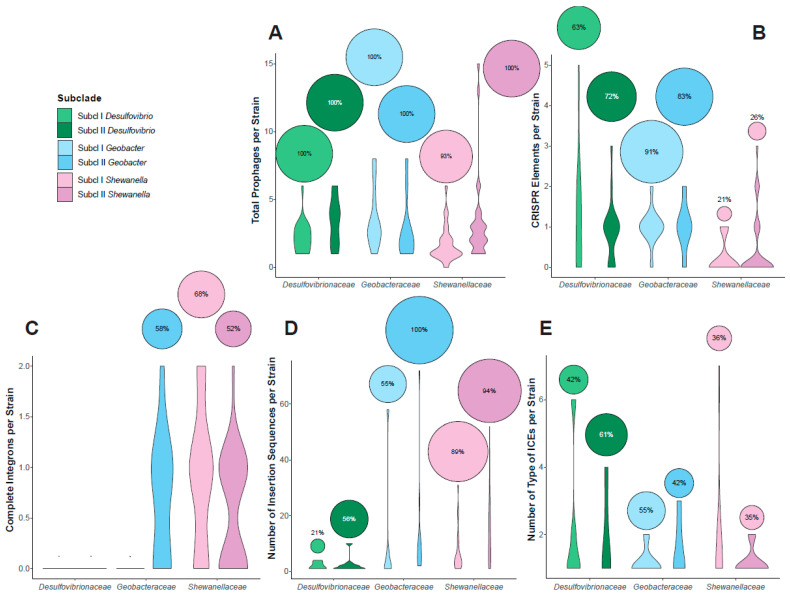
Violin plots representing the distribution of elements associated with the mobilome across different subclades and families; (**A**) total prophages per strain, (**B**) CRISPR elements per strain, (**C**) complete integrons per strain, (**D**) number of insertion sequences per strain, and (**E**) number of types of ICEs per strain. The circles on the top represent the prevalence of each element found in each family. Dots on the top represent that no strain was found to contain integrons.

**Figure 7 microorganisms-12-01796-f007:**
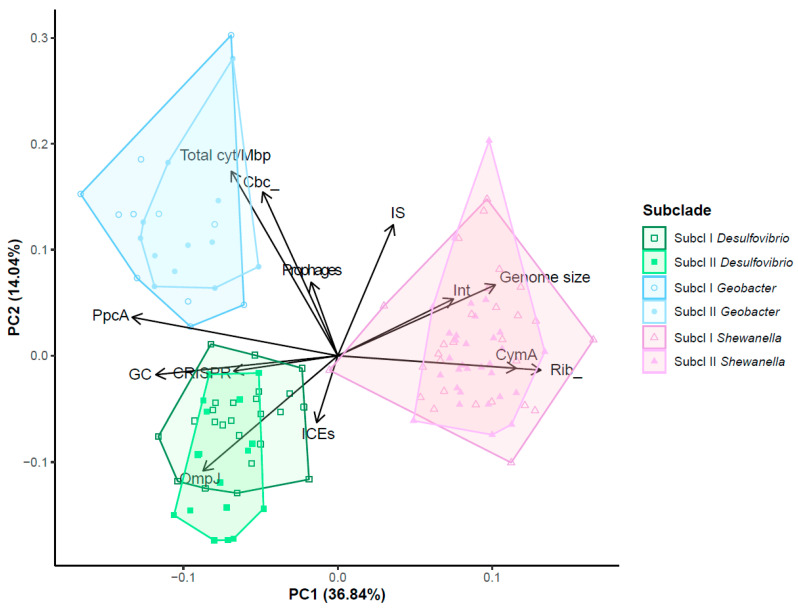
Principal Component Analysis of the strains from the *Desulfovibrionaceae*, *Geobacteraceae*, and *Shewanellaceae* families in relation to their mobilome components, types of cytochromes, genome size, and GC content. The strains were compared with the following parameters: genome size (Genome_size), GC% (GC), total prophages per strain (Prophages), total cytochromes per Mbp (Total cyt/Mbp), complete integrons per genome (Int), CRISPRs (CRISPR), number of insertion sequences per strain (IS), types of ICEs per strain (ICEs), number of copies of PpcA per strain (PpcA), number of copies of OmpJ per strain (OmpJ), sum of the number of copies of Cbc (Cbc_), number of copies of CymA (CymA), and sum of the number of copies of Rib (Rib_). The shapes indicate which family the strain is from, and the filling/color indicates the corresponding subcluster.

**Figure 8 microorganisms-12-01796-f008:**
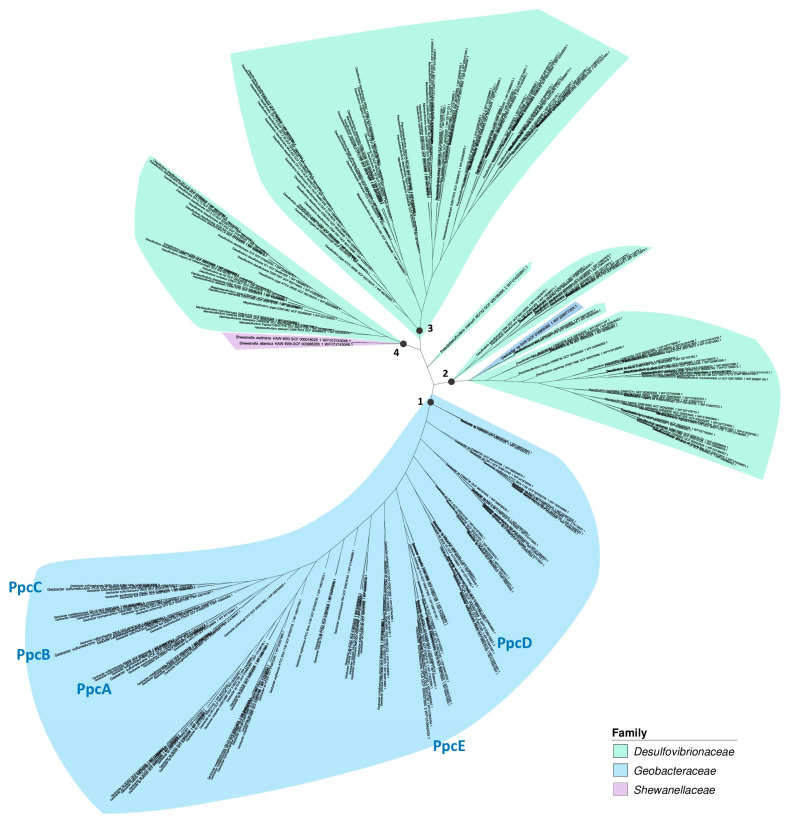
Phylogenetic tree of proteins homologous to PpcA. The phylogenetic tree was inferred using Orthofinder v2.5.4, and corresponds to the orthologous group where the periplasmic triheme cytochrome PpcA is present. The tree is composed of 257 protein sequences: 112 from *Geobacter* strains, 2 from *Shewanella*, and 143 from members of the *Desulfovibrionaceae* family. Each clade/branch is colored according to its family of origin. In the figure, the positions of PpcA and its homologs PpcB, PpcC, PpcD, and PpcE in *G. sulfurreducens* PCA are indicated.

**Figure 9 microorganisms-12-01796-f009:**
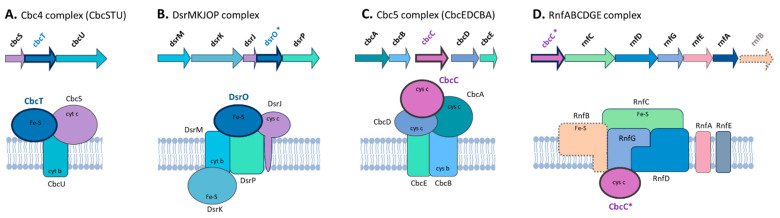
Schematic representation of membrane-bound complexes Cbc4, Dsr, Cbc5 and Rnf involved in EET. Gene contexts and depictions of membrane-associated complexes: (**A**) The Cbc4 complex (cbcSTU) identified in *G. sulfurreducens* PCA, consists of three subunits: a *b*-type cytochrome CbcU, an iron–sulfur cluster-binding protein CbcT, and a *c*-type cytochrome CbcS; (**B**) The Dsr complex (DsrMKJOP) in family *Desulfovibrionaceae* members, comprised of five subunits: a *b*-type cytochrome DsrM, an iron-sulfur binding protein DsrK; a triheme *c*-type cytochrome DsrJ; a ferredoxin-like protein DsrO (homologous to CbcT), and an NrfD/PsrC family integral membrane protein DsrP. The orthologous proteins CbcT from the Cbc4 complex and DsrO from the Dsr complex are highlighted in blue. (**C**) The Cbc5 complex (cbcEDCBA) from *Geobacter* species, composed of five subunits: three *c*-type cytochromes CbcA, CbcC and CbcD, a *b*-type cytochrome CbcB, and a membrane protein CbcE; and (**D**) The Rnf complex (rnfABCDGE) consists of six subunits, including four membrane proteins RnfA, RnfD, RnfE, and RnfG and two iron-sulfur proteins, RnfB and RnfC. The protein homologous to CbcC (CbcC*) is shown on the cytoplasmic side, which is possibly recruited by the Rnf complex for its function. The orthologous proteins CbcC from the Cbc5 complex and Cbc* from the Rnf complex are highlighted in violet. The rnfB subunit is indicated with dashed lines to denote its variable presence in the Rnf operon of *Desulfovibrionaceae* family. The complexes are depicted within the inner membrane, with the periplasm located at the top and the cytoplasm at the bottom.

## Data Availability

Data are contained within the article.

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
