# Peer review of "Novel Insights on Extracellular Electron Transfer Networks in the Desulfovibrionaceae Family: Unveiling the Potential Significance of Horizontal Gene Transfer"

_microorganisms, 2024, doi:10.3390/microorganisms12091796_

Round 1
Reviewer 1 Report
Comments and Suggestions for Authors
Authors here present an article analyzing EET genes/proteins from Desulfovibrionaceae and compared them with two popularly explored EAB. The study is an interesring investigation and excellent article for a beginer researcher on Electromicrobiology/BES. They have done a genomic analysis applying some really good and curent available bioinformatics tools. It might still need a few corrections further before being considered for publication.
It is not mentioned that the genomes were sequenced from their laboratory or obtained from the database. Please make it clear and mention the availability of genomes information.
Authors mention about the CHECKM2 did they run binning for sequences prior to doing quality check?
Figure 4, please look at correcting the spellcheck marks underlined in red.
The studies report a very good genome analysis of Desulfovibrionaceae supported with wonderful supplimentary informaion.
Author Response
Reviewer #1
- Authors here present an article analyzing EET genes/proteins from Desulfovibrionaceae and compared them with two popularly explored EAB. The study is an interesring investigation and excellent article for a beginer researcher on Electromicrobiology/BES. They have done a genomic analysis applying some really good and curent available bioinformatics tools. It might still need a few corrections further before being considered for publication.
We really appreciated this comment because it reflects how our lab is moving towards exploring the field of electromicrobiology. Therefore, we hope this piece of research may help others who are moving in the same direction.
- It is not mentioned that the genomes were sequenced from their laboratory or obtained from the database. Please make it clear and mention the availability of genomes information.
The manuscript mentioned this in the methodology, "Sequence data for all of the bacterial genomes were retrieved from the NCBI RefSeq database (query date: March 2022)."
However, to emphasize that those genomes were available in databases and not sequenced by our group, we changed this “A representative number of genomes from both sulfate-reducing bacteria (SRB) and Fe-reducing bacteria (FeRB) were analyzed to investigate the shared genomic characteristics and the evolutionary relationships between the two groups” for “A collection of already-sequenced genomes from both sulfate-reducing bacteria (SRB) and Fe-reducing bacteria (FeRB) were analyzed to investigate the shared genomic characteristics and the evolutionary relationships between the two groups.”
- Authors mention about the CHECKM2 did they run binning for sequences prior to doing quality check?
Since we intend to use reliable genomic information available, we used CheckM2 exclusively to check the quality of the genomes. As a consequence, we did not perform binning.
- Figure 4, please look at correcting the spellcheck marks underlined in red.
Thanks for the comment. This issue is fixed in the new version of the manuscript.
- The studies report a very good genome analysis of Desulfovibrionaceae supported with wonderful supplimentary informaion.
We appreciate this comment. We tried to combine comparative genomic analysis with the phenotypic information available for SRB. Since this information is rather spaced and disjointed, we invested time in reconstructing a solid database that will be available for the readers.

Reviewer 2 Report
Comments and Suggestions for Authors
Ovrall, I found the manuscript ofgreat value however as i mentioned earlier, majority of refernces are too old.
Author Response
Reviewer #2
- Ovrall, I found the manuscript of great value however as i mentioned earlier, majority of refernces are too old.
We thank the reviewer for this comment. However, since we integrated phenotypic information available for SRB, this is mainly available from publications of 1980´s -2000´s, when the approach of isolation and characterization of SRB was highly productive. To improve this aspect, we added the following more recent references, all of them has been highlighted in yellow in the text:
Fike, D. A., A. S. Bradley and C. V. Rose (2015). "Rethinking the Ancient Sulfur Cycle." Annual Review of Earth and Planetary Sciences 43(Volume 43, 2015): 593-622
Kirk, M. F., Q. Jin and B. R. Haller (2016). "Broad-Scale Evidence That pH Influences the Balance Between Microbial Iron and Sulfate Reduction." Groundwater 54(3): 406-413.
Fu, H., M. Jin, L. Ju, Y. Mao and H. Gao (2014). "Evidence for function overlapping of
CymA and the cytochrome 1 complex in the hewanella oneidensis nitrate and nitrite
respiration." Environmental Microbiology 16(10): 3181-3195.
Heitmann, T. and C. Blodau (2006). "Oxidation and incorporation of hydrogen sulfide by dissolved organic matter." Chemical Geology 235(1): 12-20.
Kappler, A., C. Bryce, M. Mansor, U. Lueder, J. M. Byrne and E. D. Swanner (2021). "An evolving view on biogeochemical cycling of iron." Nature Reviews Microbiology 19(6): 360-374.
Yu, Z.-G., S. Peiffer, J. Göttlicher and K.-H. Knorr (2015). "Electron Transfer Budgets and Kinetics of Abiotic Oxidation and Incorporation of Aqueous Sulfide by Dissolved Organic Matter." Environmental Science & Technology 49(9): 5441-5449.

Reviewer 3 Report
Comments and Suggestions for Authors
The reviewed manuscript, microorganisms-3094580, is dedicated to the understanding of the distribution of proteins related to extracellular electron transfer processes and their role in iron and sulfur biogeochemical cycles. The presented study is a good experimental work that contains original results and, I believe, will be interesting to readers.
I recommend the manuscript be published in Microorganisms after minor revision.
My comments are related to the preparation of the manuscript:
1. The abstract must be reduced (see Instructions for Authors).
2. "Fe(III)" is mentioned twice in one sentence in line 20.
3. The text in Figs. 1, 5, and 8 is hardly readable. Please increase the size if possible.
Author Response
Reviewer # 3
- The reviewed manuscript, microorganisms-3094580, is dedicated to the understanding of the distribution of proteins related to extracellular electron transfer processes and their role in iron and sulfur biogeochemical cycles. The presented study is a good experimental work that contains original results and, I believe, will be interesting to readers.
- The abstract must be reduced (see Instructions for Authors).
We thank the reviewer for this comment since we did not realize that the previous abstract exceeded the word limit. The current version of the abstract meets the requirements and standards stated in the instructions.
- "Fe(III)" is mentioned twice in one sentence in line 20.
This is fixed in the new version of the manuscript.
- The text in Figs. 1, 5, and 8 is hardly readable. Please increase the size if possible.
We thank the reviewer for this comment. We improved Figures 1, 5, and 8 by modifying the font size of the species names included and increasing their resolution. Changing the species names by only protein ID did not improve the legibility of the phylogenetic tree (Figure 8).
